# EXPLORE TO MIMIC: A REINFORCEMENT LEARNING BASED AGENT TO GENERATE ONLINE SIGNATURES

## ABSTRACT

Recent advancements in utilising decision making capability of Reinforcement Learning (RL) have paved the way for innovative approaches in data generation. This research explores the application of model free on-policy RL algorithms for generating online signatures and its controlled variations. Online signatures are captured via e-pads as sequential structural coordinates. In this study, we have introduced a robust on-policy RL agent named as SIGN-Agent, capable of generating online signatures accurately. Unlike other RL algorithms, on-policy RL directly learns from the agent's current policy, offering significant advantages in stability and faster convergence for sequential decision-making. The proposed SIGN-Agent operates in a random continuous action space with controlled exploration limits, allowing it to capture complex signature patterns while minimizing errors over time. The downstream applications of this system can be extended in diverse fields such as enhancing the robustness of signature authentication systems, supporting robotics, and even diagnosing neurological disorders. By generating reliable, human-like online signatures, our approach strengthens signature authentication systems by reducing susceptibility towards system-generated forgeries, if trained against them. Additionally, the proposed work is optimized for low-footprint edge devices, enabling it to function efficiently in the area of robotics for online signature generation tasks. Experimental results, tested on large, publicly available datasets, demonstrate the effectiveness of model free on-policy RL algorithms in generating online signature trajectories, that closely resemble user's reference signatures. Our approach highlights the potential of model free on-policy RL as an advancement in the field of data generation targeting the domain of online signatures in this research.

## 1 INTRODUCTION

Signatures are a widely recognized biometric tool for verifying an individual's identity. The inherent complexity and uniqueness of signatures have always attracted researchers aiming to develop advanced authentication systems. With the rise of digital platforms and devices, online signatures, captured through e-pads, have gained significant attention. These signatures capture both the structural and behavioral characteristics of an individual, making them highly valuable for secure authentication. Typically, authentication systems are trained on large datasets, where signature forgeries are manually generated by imitating genuine signatures. However, as these systems rely on human ability of mimicking, hence generating the need of having sophisticated online signature generation system to make authentication methods robust against digitally generated forgeries as generated features are always a subset of the distinguishing features Tamaazousti et al. (2017).

Furthermore, the potential uses of this technology extend beyond mere convenience, finding relevance in critical domains like finance, legal affairs, and healthcare (Bibi et al., 2020). Application of signature generation can be utilized for numerous downstream tasks, along-with making robust authentication system (Pandey et al., 2024). In the realm of robotics, our proposed agent enables robots to generate human-like signatures and can be extended to handwriting with a high degree of accuracy and natural flow (Zhao et al., 2020). This capability could enhance human-robot interaction, where robots are equipped with performing tasks that require fine motor skills. Additionally, proposed agent has significant potential in diagnosing neurological conditions such as Parkinson's, Alzheimer's, and dyslexia by analyzing signature and handwriting trajectories (Gornale et al., 2022).

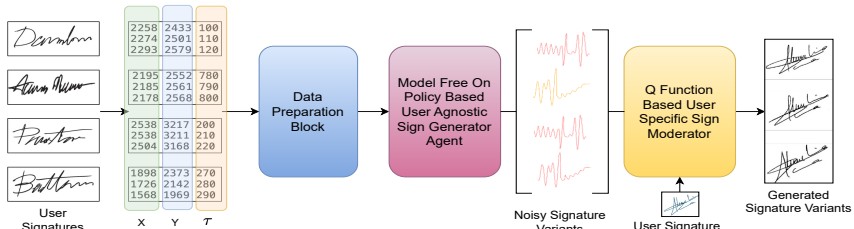

Figure 1: High level flow demonstrating signature generation including role of model free on policy based sequential decision making SIGN–Agent and Sign Moderator block.

Modeling these unique signature trajectories also holds great importance for forensic applications (Khan et al., 2023).

Online signatures consist of continuous time-series data, specifically Cartesian coordinates $(x, y)$, weighted by pressure $(p)$, and sampled at regular intervals $(\tau)$. Generating this data presents a unique challenge due to the variability and randomness inherent in each individual's signing behavior. Although prior work has tackled time-series data generation in domains like forecasting and random masking, but the problem of generating realistic online signatures remains under-explored. The randomness in signing patterns introduces a level of difficulty not present in simpler time-series tasks. In our pursuit of developing an efficient model for online signature generation, we initially explored established generative techniques such as Transformer networks (Zhu & Soricut, 2021), Generative Adversarial Networks (GANs) (Smith & Smith, 2020), and Diffusion models (Alcaraz & Strodthoff, 2022). While these approaches demonstrated promising results, they exhibited certain limitations. These included difficulties in capturing long-range dependencies and challenges related to computational costs and training stability (Smith & Smith, 2020). A comparative performance analysis of these methods is presented in Table 8.

In this research, we tackle the challenge of online signature generation by utilizing model free on-policy Reinforcement Learning (RL) algorithms namely Proximal Policy Optimization (PPO), Trust Region Policy Optimization (TRPO), and Advantage Actor-Critic (A2C) using SIGN-Agent. These algorithms are particularly suited to tasks requiring sequential decision-making, such as signature generation, due to their stable training dynamics and efficient policy optimization. Unlike off-policy methods that rely on past experiences, on-policy RL continuously updates its policy based on real-time interactions with environment, making it more adaptive to the variability of human signatures. Our proposed method trains the agent to learn the underlying distribution of $x$ and $y$ coordinates in an online signature, with the action space designed as random continuous with defined limits to allow precise replication of stroke dynamics. Controlled exploration is achieved by introducing stochastic noise into each action, allowing the agent to capture individual variations in signing patterns without deviating from the core structure. During inference, a noise variance $(NV)$ is applied to the generated $x$ and $y$ coordinates to simulate natural variability in signatures. We have trained and tested this approach with sequential as well as non-sequential network architectures as part of policy networks. Additionally, a Sign Moderator block (SM), based on a learned Q-function, is introduced to select the best normalized coordinates that align with the user-specific signature distribution. Experimental results highlight the effectiveness of PPO, TRPO, and A2C in producing high-quality signatures, where PPO is slightly better in producing higly resembling signatures because of stability in learning.

Given method shows the real time performance $(UserSigningTime \approx SignGenerationTime)$ on small edge hardware like Raspberry Pi, making it an adequate candidate for environment friendly system as well as for robotic applications. To the best of our knowledge, SIGN-Agent represents the first framework explicitly developed for online signature generation. Unlike previous works that treat signatures as generic time-series data, SIGN-Agent models them as intricate, user-defined temporal sequences, addressing both the spatial and dynamic complexities unique to this domain. Figure 1, present the high level block diagram, demonstrating model free on-policy RL models in generating high-quality, realistic online signatures. The main contributions of this paper are as follows:

Table 1: Comparison of Prior Approaches, their respective limitations as quoted in papers and its comparison with SIGN-Agent.

| Category | Approach | Limitations | How SIGN-Agent Differs |
|---|---|---|---|
| Traditional Models | HMMs (Rúa & Castro, 2012) | Poor generalization to user variability | Dynamically adapts to diverse user-specific patterns. |
| Generative Models | GANs (Goodfellow et al., 2014) | Training instability; mode collapse | Ensures stability and consistency via on-policy RL |
| | VAEs (Tolosana et al., 2021) | Overly smooth outputs; lacks fine details | Preserves fine-grained signature dynamics. |
| | Diffusion Models (Alcaraz & Strodthoff, 2022) | High computational cost; unsuitable for real-time applications | Optimized for real-time, low-latency generation. |
| Imitation Learning (IL) | Behavior Cloning (Pomerleau, 1991) | Compounding errors; lacks robustness | Handles variability with dynamic RL-based adjustments. |
| Reinforcement Learning | PPO (Schulman et al., 2017) | Data inefficiency; requires stable policy updates | Balances stability and exploration with efficient on-policy updates. |
| | TRPO (Schulman, 2015) | Computationally expensive for large-scale tasks | Ensures computational efficiency with adaptive trust region updates. |
| | A2C (Mnih, 2016) | Limited scalability; struggles with user-specific refinement | Combines fast convergence with user-specific trajectory adjustments. |

- We propose the formulation of online signature generation as a model-free on-policy RL agent, using Q-Learning-based Sign Moderator for enhanced sequential decision-making.

- An optimized agent for low-footprint devices utilising sequential networks as RL policy with futuristic reward mechanism for effective long-range signature trajectory generation.

## 2 RELATED WORK

The generation of realistic online signatures has garnered significant research interest due to its applications in biometric authentication and secure identity verification. Traditional methods like Hidden Markov Models (HMMs) (Rúa & Castro, 2012) were foundational in capturing temporal dependencies in signature sequences. However, their sensitivity to variations and limited generalization hinder their real-world applicability.

Recent studies have explored modern generative models for signature generation. Transformers (Vaswani et al., 2017), adapted for handwriting tasks (Li et al., 2021), excel at modeling long-range dependencies but rely on computationally intensive techniques like $top_k$ sampling, which limits their ability to capture continuous, user-specific signature dynamics. GANs (Goodfellow et al., 2014), popular for handwriting synthesis (Zhang et al., 2019; Alonso-Fernandez et al., 2019), often suffer from training instability, leading to inconsistent user-specific outputs. VAEs (Kingma & Welling, 2013), with their latent space representations, enable controlled variation but struggle to capture the fine-grained details essential for realistic signature replication. Diffusion Models (Alcaraz & Strodthoff, 2022), while producing high-quality outputs, are computationally expensive and less suited for real-time applications. Our work addresses these challenges by leveraging RL, which dynamically adapts to user-specific variability without relying on handcrafted features or extensive tuning.

Imitation Learning (IL) approaches, such as Behavior Cloning (Pomerleau, 1991) and GAIL (Ho & Ermon, 2016), have shown promise in mimicking human actions. However, IL methods are prone to compounding errors and policy drift, making them less reliable in tasks with high variability, such as user-specific signature generation. Unlike IL, which relies heavily on expert demonstrations, our RL-based approach balances exploration and exploitation, enabling the model to adapt dynamically to diverse user trajectories and generate robust, personalized signatures.

Reinforcement Learning (RL) offers a robust alternative for tasks requiring sequential decision-making and adaptability. Model-free RL algorithms like Proximal Policy Optimization (PPO) (Schulman et al., 2017), Trust Region Policy Optimization (TRPO) (Schulman, 2015), and Advantage Actor-Critic (A2C) (Mnih, 2016) are particularly suited for dynamic environments. Unlike generative models, RL methods dynamically balance exploration and exploitation, making them highly effective for modeling the variability and complexity of online signatures. While RL has not been widely applied to online signature generation, our work leverages on-policy RL to train SIGN-Agent, allowing it to generalize across users and adapt dynamically to their unique signature trajectories. The integration of a Q-learning-based Sign Moderator ensures further refinement of user-specific dynamics, addressing the limitations of prior RL methods.

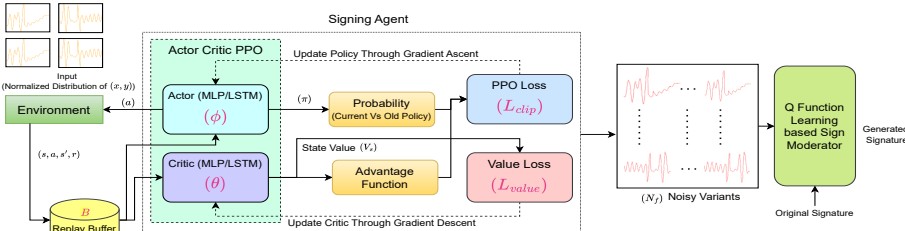

Figure 2: Architecture diagram of sequential decision making SIGN-Agent comprising of PPO Actor Critic and Sign Moderator

Our proposed SIGN-Agent introduces a two-phase RL-based framework tailored explicitly for online signature generation. In the first phase, the agent learns a foundational "scribble" structure to approximate general signature dynamics. In the second phase, a Q-learning-based Sign Moderator (SM) refines these dynamics to match individual user patterns. This dynamic adjustment allows SIGN-Agent to generate realistic, user-specific signatures without extensive tuning or reliance on handcrafted features. To our knowledge, SIGN-Agent is the first framework designed explicitly for online signature generation, advancing the field by treating signatures as intricate, user-defined temporal sequences rather than generic time-series data.

## 3 METHODOLOGY

This section details the modeling of the RL based SIGN-Agent, designed for generating online signatures, as illustrated in Figure 2. Illustration on the problem formulation with its associated challenges and solution methodology is also given in this section.

### 3.1 OVERVIEW OF PROPOSED METHOD

The SIGN-Agent leverages three on-policy reinforcement learning (RL) algorithms—Proximal Policy Optimization (PPO), Trust Region Policy Optimization (TRPO), and Advantage Actor-Critic (A2C)—to address the challenges of generating realistic and user-specific online signatures. Each algorithm brings complementary strengths that align with the requirements of this task, such as stability, adaptability, and efficient convergence. **PPO:** Ensures robust policy updates by balancing exploration and exploitation, making it effective in noisy environments. PPO's stability is particularly valuable in training the SIGN-Agent on diverse user-specific trajectories (De La Fuente & Guerra, 2024). **TRPO:** Provides smooth trajectory generation by constraining policy updates within a trust region. This enhances precision, ensuring smoother transitions between consecutive points in the signature trajectory Shani et al. (2020). **A2C:** Accelerates convergence through parallelized actor-critic updates, enabling efficient learning across diverse signature patterns. A2C is especially useful for exploring a wide range of variations during training (Gerpott et al., 2022). The inclusion of all three algorithms is motivated by their complementary strengths. Empirical results in Table 3, 5, 4 demonstrate their unique contributions, with PPO excelling in stability, TRPO producing smoother trajectories, and A2C achieving faster convergence. These experimental results also validate the strengths of each algorithm. PPO demonstrates superior stability, as reflected in its lower variance in KLD and MSE metrics during training as shown in Table 4. TRPO generates smoother signature trajectories, evident from its higher cosine similarity scores when compared to target trajectories. A2C achieves faster convergence, reducing training iterations by approximately 20% compared to PPO and TRPO, though it exhibits slightly higher variance in signature fidelity. These observations justify the inclusion of all three algorithms within SIGN-Agent. The decision to use PPO, TRPO, and A2C stems from their complementary characteristics in addressing the unique challenges of online signature generation: PPO ensures stable and robust training by clipping probability ratios during updates, reducing the likelihood of policy divergence. TRPO maintains precision by constraining updates within a trust region, enabling the generation of smooth and realistic signature trajectories. A2C accelerates convergence through parallelized updates, facilitating efficient exploration of diverse signature patterns. SIGN-Agent balances stability, adaptability, and efficiency, as evidenced by experimental results in Table 3, 5, 4. Ablation studies (Table 4) further highlight the

impact of each algorithm, with PPO and TRPO excelling in fidelity metrics, while A2C improves training efficiency. As the agent is trained over a distribution of user-specific signature data with a limited number of initial points ($\leq 20$) and noise variation factor (NV), it produces signature variations in a robust, user-agnostic manner. A detailed mathematical formulation for each algorithm is provided in Appendix A.

## 3.2 RL PROBLEM FORMULATION

The RL-based SIGN-Agent is formulated as a sequential decision-making task to generate realistic online signatures. This section provides a detailed explanation of the agent, environment, state and action dimensions, policy architecture, reward function, and termination mechanism.

**Neural Network Policy Architecture:** The policy network is an LSTM-based neural network designed to capture the temporal dependencies inherent in signature trajectories. It employs a Long Short-Term Memory (LSTM) layer with a hidden size of 50, which processes input sequences with a single feature dimension (input_size = 1), representing either $x$ or $y$ coordinates of the trajectory. The LSTM sequentially processes the input data and generates hidden states at each time step. The final hidden state is passed through a fully connected linear layer that maps the features to the desired output dimension (output_size = 1), predicting the next $x$ or $y$ coordinate. Hidden and cell states are initialized to zeros to ensure compatibility with gradient tracking and device execution. This architecture is optimized for time-series prediction tasks, leveraging historical patterns to predict the next trajectory point with high accuracy.

**State and Action Dimensions:** The state $s_t$ is represented using a sliding window mechanism, capturing recent trajectory points and encapsulating temporal dependencies in the signature generation process. Mathematically, the state is defined as $s_t = [x_{t-w}, y_{t-w}, \ldots, x_t, y_t]$, where $w$ represents the fixed window size. This representation provides the agent with sufficient historical context for predicting the next trajectory point. The action $a_t$ corresponds to the predicted next trajectory point, defined as $a_t = [x_{t+1}, y_{t+1}]$, and is sampled from a continuous action space.

**Environment Determinism:** The environment for the SIGN-Agent is deterministic, with state transitions solely dependent on the sliding window of recent trajectory points. The next state $s_{t+1}$ is determined by the transition function $s_{t+1} = f(s_t, a_t)$, where $f(\cdot)$ appends the agent's predicted action to the sliding window. Although the environment is deterministic, stochasticity is introduced during training by perturbing the agent's actions with Gaussian noise. The perturbed action is defined as $a_t = a_t' + \epsilon$, where $\epsilon \sim \mathcal{N}(0, \sigma)$. This noise simulates variability in human signature trajectories, enhancing the model's ability to generalize across diverse signature styles.

**Capturing Multiple Signature Styles:** The policy captures variations in signature styles by training on diverse user-specific datasets that include a wide range of signature patterns. The state representation integrates historical trajectory points from the current signature, latent features encoding user-specific style attributes, and global training data encompassing all signature variations for each individual. This comprehensive representation enables the policy to generalize across a variety of styles while dynamically adapting to specific trajectories during inference.

**Reward Function:** The reward mechanism plays a critical role in guiding the agent to produce user-specific signature trajectories. At each time step $t$, the reward $r_t$ is computed as the negative Euclidean distance between the generated and target points as shown in equation 1:

$$r_t = -\|(x_t, y_t) - (x_t^{\text{target}}, y_t^{\text{target}})\| \tag{1}$$

where $(x_t, y_t)$ represents the generated point, and $(x_t^{\text{target}}, y_t^{\text{target}})$ represents the corresponding target point. To ensure scale invariance, the input coordinates are normalized before computing the reward. Although no Gaussian kernel is applied, the point-wise nature of the reward focuses the agent on fine-grained accuracy during training.

**Termination Mechanism:** The generation process terminates based on a combination of two mechanisms. First, a predefined maximum trajectory length ensures that the model generates signatures within practical bounds. Second, a dynamic stopping condition is incorporated, relying on zero-pressure signals from the input data. When the pen tip is lifted off the digital pad, the system recognizes this as a termination signal, effectively mimicking the end of a user's signature. These mechanisms ensure real-world writing behaviors, accommodating variations in signature strokes and styles.

**Integration of Components:** By combining an LSTM-based policy network, a deterministic environment, and a reward-driven optimization strategy, SIGN-Agent dynamically adapts to user-specific trajectories. The framework leverages historical trajectory data, Gaussian noise for vari-

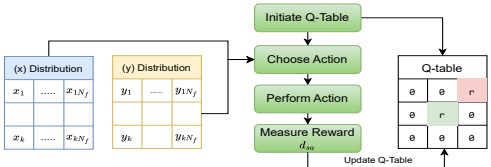

Figure 3: Illustration of Sign Moderator block working, utilizing Q-table learning through actions chosen from $x$ and $y$ noisy distributions

ability, and robust termination conditions to produce accurate and realistic signature trajectories. This integration ensures stability, precision, and adaptability, making SIGN-Agent effective across diverse signature styles and user requirements.

### 3.3 ROLE OF SIGN MODERATOR (SM)

The Sign Moderator (SM) is a critical component designed to refine the trajectory outputs of the SIGN-Agent. It operates as a post-processing step that integrates noisy variations generated by the RL policy network and produces a clean and unified signature trajectory. The SM is based on a Q-learning framework, leveraging a Q-table to select optimal trajectory points and enhance output consistency.

**Purpose and Scope:** The primary function of the SM is to smooth and refine the noisy signature trajectories produced by the RL network. While the RL policy generates multiple variations for each axis $(x, y)$ of a single signature, the SM integrates these variations to reconstruct a trajectory that closely resembles the target signature. The SM is applied during both training and inference phases to maintain consistency and ensure robust trajectory generation.

**Q-Table Construction:** The Q-table in the SM is a matrix where rows correspond to temporal states (time steps) and columns represent the available candidate trajectory points generated for each coordinate. Each entry in the Q-table, denoted as $Q(s_t, a_t)$, stores the expected cumulative reward for selecting a specific trajectory point $a_t$ at state $s_t$. The reward function aligns with the trajectory refinement goal, favoring points that minimize discrepancies between generated and target signatures.

**Q-Learning Process:** The SM employs Q-learning to iteratively update the Q-table based on the observed rewards. The Q-value updates are governed by the Bellman equation 2:

$$Q(s_t, a_t) \leftarrow Q(s_t, a_t) + \alpha \left[ r_t + \gamma \max_{a'} Q(s_{t+1}, a') - Q(s_t, a_t) \right] \tag{2}$$

where $\alpha$ is the learning rate, $\gamma$ is the discount factor, $r_t$ is the immediate reward for selecting $a_t$, and $\max_{a'} Q(s_{t+1}, a')$ represents the maximum future reward for the next state $s_{t+1}$. This iterative process enables the SM to learn optimal trajectory refinements dynamically.

**Planning and Execution:** Planning in the SM involves evaluating all candidate trajectory points at each time step to identify the one that maximizes the Q-value. This decision-making process is repeated sequentially across the trajectory, ensuring smooth transitions and alignment with user-specific patterns. By iteratively refining the trajectory, the SM minimizes noise and ensures that the generated signature adheres to structural and temporal constraints.

**Training and Inference Phases:** During training, the SM operates in conjunction with the RL policy to refine trajectories, providing feedback that improves the overall policy network. during inference, the SIGN-Agent operates without requiring re-training or re- learning for specific users. Instead, the agent takes an initial set of points from the target signature and generates multiple signature trajectories based on its trained policy. These trajectories are then refined by a Q-learning-based SM, which adjusts the output to ensure alignment with user-specific characteristics.

**Integration with RL Policy:** The SM seamlessly integrates with the RL policy network, enhancing the fidelity of generated trajectories. By selecting optimal trajectory points through Q-learning, the SM bridges the gap between noisy intermediate outputs and high-quality final trajectories, ensuring

Table 2: Publicly available online signature datasets captured on various equipment makers' digital e-pads using stylus.

| S.No | Participating Datasets | Users | Acquisition Device | Sign/User |
|------|------------------------|-------|--------------------|-----------|
| 1 | MYCT | 330 | Wacom, Intuos A6 | 25 |
| 2 | Biosecure-ID | 400 | Wacom 3 | 16 |

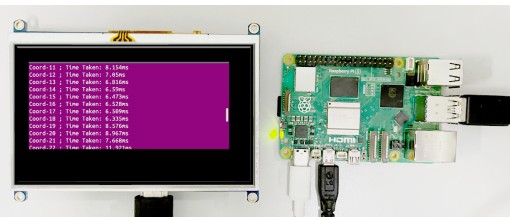

Figure 4: Inference on Raspberry Pie for SIGN-Agent, showing real time performance of signature generation

robustness and user-specific accuracy.

Ablation results on SM block given in Table 7, demonstrate that incorporating the SM significantly reduces noise and improves similarity metrics, between generated and target signatures.

## 4 EXPERIMENTAL ANALYSIS

Evaluating the quality of the generated signatures is essential for substantiating the model's performance. To achieve this, we analyzed both the generated signatures and the original signature data using a variety of similarity metrics.

### 4.1 DATA PREPARATION

In this study, two publicly available online signature datasets, MCYT (Ortega-Garcia et al., 2003) and Biosecure-ID (Fierrez et al., 2010), are utilized, as summarized in Table 2. These datasets provide significant intra-user variance by capturing signatures across multiple sessions over time and utilizing various devices, thereby ensuring adequate variability. The online signature data includes the $x$ and $y$ coordinates, along with timestamps for each recorded coordinate. Initially, the data is **standardized** by subtracting the mean $\mu$ from each coordinate and then dividing by the variance $\sigma$ to achieve scale invariance (Rutkowski & Svetina, 2014). In addition to standardization, all signatures are adjusted to a consistent length to account for variability in the dataset. Shorter signatures are extended using polynomial interpolation to generate smooth intermediate points. This preprocessing ensures uniformity during training.

During evaluation, to handle temporal misalignment between the generated and target signatures, we employ dynamic time warping (DTW). DTW aligns the sequences by stretching or compressing segments, minimizing temporal distance and enabling accurate comparison. Subsequently, **min-max normalization** is applied to prepare the data for training. The equations for standardization and normalization (Tolosana et al., 2015) are given below as Eqn. 3:

$$(c_i) = \frac{C - \mu}{\sigma}, \quad c'_i = \frac{c_i - c_{min}}{c_{max} - c_{min}} \tag{3}$$

where $(c_i)$ represents standardization, $c'_i$ denotes normalization, and $C$ refers to the coordinate distribution defined as $C = c_1, c_2, c_3, \ldots, c_n$.

### 4.2 EXPERIMENTAL DETAILS

The proposed model-free on-policy RL SIGN-Agent is designed for computational efficiency, significantly reducing computational overhead compared to traditional RL approaches. The architecture

Table 3: Average Signature Generation time and Actual Signature Elapsed Time comparison

| Dataset | Processor | Generation Time (sec) | Elapsed Time (sec) | CPU Frequency(GHz) |
|---------|-----------|----------------------|--------------------|--------------------|
| MCYT | Intel i7 | 2.4109 | 2.9937 | 4.9 |
| | Intel i5 | 2.9543 | 2.9937 | 3.4 |
| | RaspPie (ARM Cortex) | 3.1432 | 2.9937 | 2.4 |

Table 4: Comparative performance evaluation using KLD, MSE and Cosine Similarity by varying $NV$ values for MCYT and Biosecure-ID datasets using PPO policy

| Metrics | | KLD | | | MSE | | | Cosine Similarity | | |
|---------|---|--------|---------|---------|--------|---------|---------|--------|---------|---------|
| Dataset | | $NV = 5$ | $NV = 10$ | $NV = 15$ | $NV = 5$ | $NV = 10$ | $NV = 15$ | $NV = 5$ | $NV = 10$ | $NV = 15$ |
| MCYT | x | 0.0802 | 0.2835 | 0.4926 | 0.0729 | 0.0901 | 0.0925 | 97.19 | 96.74 | 94.03 |
| | y | 0.0693 | 0.1941 | 0.2863 | 0.0845 | 0.0845 | 0.0935 | 97.06 | 95.89 | 94.04 |
| Biosecure-ID | x | 0.8190 | 1.0920 | 1.7436 | 0.1394 | 0.3048 | 0.4903 | 96.99 | 95.72 | 93.97 |
| | y | 0.5831 | 0.8356 | 1.0958 | 0.1309 | 0.2398 | 0.4991 | 96.59 | 95.06 | 92.63 |

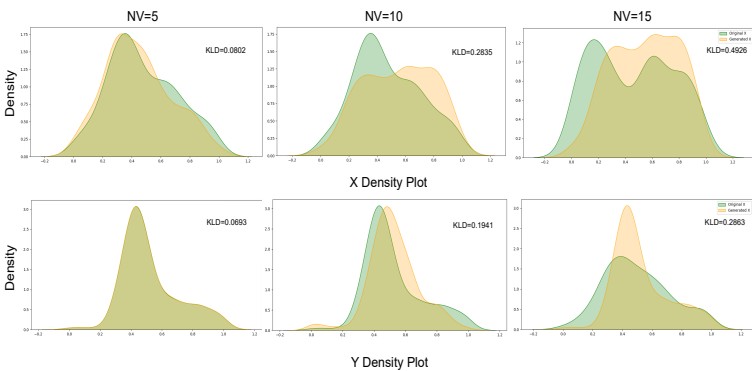

Figure 5: Distribution plot with KLD values between original and generated $x$ and $y$ coordinates of signature across varying $NV$

Table 5: Comparative performance analysis of on-policy algorithms for variation of $NV$ values using MSE for distribution of the MCYT and BioSecure-ID datasets

| Algorithm | Coordinate | MSE | | |
|-----------|------------|--------|---------|---------|
| | | $NV = 5$ | $NV = 10$ | $NV = 15$ |
| PPO | X | 0.0729 | 0.0901 | 0.0925 |
| | Y | 0.0879 | 0.0845 | 0.0935 |
| TRPO | X | 0.0859 | 0.1247 | 0.1384 |
| | Y | 0.0878 | 0.0973 | 0.1829 |
| A2C | X | 0.0925 | 0.1895 | 0.2206 |
| | Y | 0.1076 | 0.1745 | 0.2473 |

is optimized for training on a Nvidia GeForce GTX 1080 Ti GPU and facilitates low-latency inference on low foot print edge boards. SIGN-Agent can be inferred on Raspberry Pi, demonstrating its capability for practical deployment in resource-constrained environments. Figure 4, illustrates the Raspberry Pi-based setup for the SIGN-Agent, demonstrating its capability to perform real-time signature generation. The display in Table 3, showcases the time taken for each signature coordinate generation, highlighting the efficiency and responsiveness of the system.

In our model-free on-policy RL SIGN-Agent, we strategically optimized hyperparameters to enhance performance across diverse scenarios. The training process was conducted over 5000 episodes, with a focus on managing temporal dependencies using a 20-episode window. Our neural network architecture featured three hidden layers consisting of 256, 300, and 400 units, along with two LSTM layers to effectively capture sequential patterns within the time-series data. The state dimension was designed to match the length of the time series, while the action space was continuous and one-dimensional.

In this on-policy methods, data was collected directly from the policy's interactions with the environment, ensuring that the learning process remained aligned with the most current policy. Updates

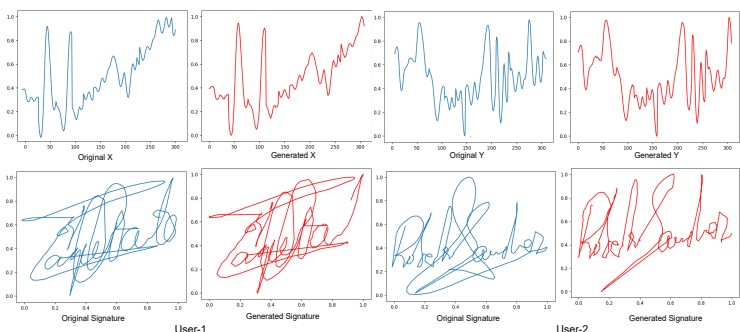

Figure 6: Illustration of actual and generated $X, Y$ coordinates and 2D-Signature through proposed SIGN-Agent

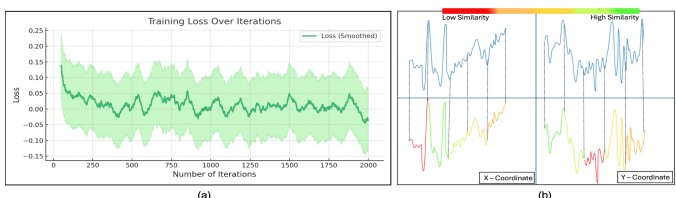

Figure 7: (a) Loss trend across training iterations for proposed SIGN-Agent (b) Similarity and Dissimilarity heat-map for generated vs original $x, y$ coordinates

to the policy network were performed using mini-batch gradient descent with a batch size of 100. To promote stable long-term learning, a discount factor of 0.99 and a smoothing coefficient of 0.95 were employed. To enhance exploration, noise was injected into the actions (0.2 policy noise, clipped at 0.5). The policy network was updated every two steps, striking a balance between exploration and exploitation. The reward and evaluation mechanisms are designed to address both length mismatches and temporal misalignment between generated and target signatures. Length mismatches are resolved by extending shorter sequences using polynomial interpolation to a consistent length. Temporal misalignment are handled through DTW, which aligns the sequences by minimizing temporal distance, ensuring a robust and fair evaluation across varying signature trajectories. Table 4, provides a comparative performance analysis using KLD, Mean Squared Error (MSE) (Hodson, 2022), and Cosine Similarity (CS) metrics across varying $NV$ values for used datasets under the PPO policy. As $NV$ increases, KLD and MSE values rise, reflecting increased divergence and error in generated signatures. However, CS remains consistently high, indicating that the structural alignment between generated and original signatures is well-preserved. During inference, the agent generates signatures from an initial set of points, and the SIGN Moderator refines them for user-specific fidelity, eliminating the need for re-learning.

**Comparative Analysis of Model-Free Algorithms** A comprehensive evaluation is conducted on model-free RL algorithms, specifically PPO, TRPO, and A2C. The performance of each algorithm is assessed by computing the MSE between the generated and original signatures. The PPO algorithm outperforms TRPO and A2C, demonstrating the best results due to its adaptive update mechanism that strikes an effective balance between exploration and exploitation. Table 5, provides a detailed comparison of the MSE results across varying $NV$ values. Figure 5, shows the plots for original and generated $x$ and $y$ coordinate distributions through PPO mentioning KLD values also for the calculated difference. **Actor-Critic Networks** We explored both sequential and non-sequential architectures for the policy networks, specifically utilizing Multi-Layer Perceptrons (MLPs) (Tang et al., 2015) and Long Short-Term Memory (LSTM) networks (Bodapati et al., 2020). Empirical evaluations as shown in Table 6 indicate that sequential architectures, such as LSTMs, exhibit a superior ability to retain long-term dependencies within signature data compared to their non-sequential counterparts. Figure 6, illustrates the actual and generated $x, y$ coordinates and 2D-signature produced by the proposed SIGN-Agent. Figure 7 (a) shows the loss trajectory, indicating the model's convergence and optimization, with smoothing applied to highlight key trends for easier

Table 6: Comparative performance analysis of MLP and LSTM networks using PPO policy across varying $NV$ values for distribution of the MCYT and BioSecure-ID datasets

| Network | Coordinate | MSE | | |
|---------|-----------|--------|---------|---------|
| | | $NV = 5$ | $NV = 10$ | $NV = 15$ |
| LSTM | X | 0.0729 | 0.0901 | 0.0925 |
| | Y | 0.0879 | 0.0845 | 0.0935 |
| MLP | X | 1.5927 | 1.7359 | 1.9284 |
| | Y | 1.4823 | 1.7004 | 1.8374 |

Table 7: Ablation study with the inclusion and exclusion of the Sign Moderator (SM) with PPO policy across varying $NV$ values for MCYT and BioSecure-ID datasets

| Ablation | Coordinate | KLD | | |
|----------|-----------|--------|---------|---------|
| | | $NV = 5$ | $NV = 10$ | $NV = 15$ |
| With SM | X | 0.0802 | 0.2835 | 0.4926 |
| | Y | 0.0693 | 0.1941 | 0.2863 |
| Without SM | X | 0.1728 | 0.4029 | 0.7391 |
| | Y | 0.1309 | 0.4017 | 0.3946 |

Table 8: Performance evaluation on state-of-the-art generative networks and SIGN-Agent using KLD in $X, Y$ and $X, Y$ direction

| Dataset | Approach | $X$ | $Y$ | $(X, Y)$ |
|---------|----------|-----|-----|----------|
| MCYT | Transformer (Zhu & Soricut, 2021) | 0.1332 | 0.7423 | 0.4176 |
| | GAN Netwrok (Smith & Smith, 2020) | 0.3814 | 0.3765 | 0.3412 |
| | Diffusion Network (Alcaraz & Strodthoff, 2022) | 0.2736 | 1.8412 | 2.0970 |
| | Proposed SIGN-Agent | 0.00237 | 0.00937 | 0.00863 |
| Biosecure ID | Transformer (Zhu & Soricut, 2021) | 0.12144 | 0.65423 | 0.46281 |
| | GAN Netwrok (Smith & Smith, 2020) | 0.6897 | 0.6981 | 0.5847 |
| | Diffusion Network (Alcaraz & Strodthoff, 2022) | 0.2638 | 0.2483 | 0.2684 |
| | Proposed SIGN-Agent | 0.002661 | 0.008374 | 0.005143 |

performance evaluation over time, the (b) part shows the heatmap between original and generated $x$ and $y$ trajectories.

**Ablation of SM Block:** In our proposed approach, we performed an ablation study on the SM block. Experiments were conducted for online signature generation both with and without the SM block. When the SM block was removed, the prediction was derived by averaging all the noisy coordinate variations to produce a single value. The results indicated that incorporating a Q-function learning-based SM block significantly enhances the generation process, improving the resemblance of the generated signatures to the original ones. Table 7, presents the results of this ablation study.

**Comparison with Other Approaches** Before selecting model-free RL algorithms for signature generation, we analyzed state-of-the-art (SOTA) generative models, including Transformers (Zhu & Soricut, 2021), Generative Adversarial Networks (GAN) (Smith & Smith, 2020), and Diffusion Models (Alcaraz & Strodthoff, 2022), which are widely applied to time-series generation tasks. Using KLD to quantify differences between the distributions of original and generated signatures, we evaluated these models on the MCYT and Biosecure-ID datasets (Table 8). While these models demonstrated some effectiveness, they are not inherently designed for the complexities of online signatures. SIGN-Agent explicitly addresses these limitations by being optimized for the unique requirements of online signature generation.

## 5 CONCLUSION

In conclusion, our study demonstrates the efficacy of the proposed SIGN-Agent for generating high-fidelity online signatures using the MCYT and Biosecure-ID datasets. By addressing inter-session variability and employing a robust on-policy optimization strategy, SIGN-Agent was able to consistently produce realistic and accurate signatures. When compared to conventional models, including transformers, GANs, and diffusion models, SIGN-Agent outperformed across diverse conditions. Furthermore, the framework was tested on low-end hardware, such as Intel i7 processors and Raspberry Pi, confirming its computational efficiency and fast inference capabilities. This makes the proposed approach cost-effective, adaptable solution for reliable online signature generation in real-world applications.

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

## A APPENDIX A: DETAILS OF THE RL ON-POLICY METHODS

Here we are giving the details of the RL on-policy Methods below:

**Proximal Policy Optimization (PPO)** In our study, we employed the PPO algorithm as a robust approach for addressing the signature generation challenge. PPO optimizes the policy using a clipped surrogate objective function, defined as:

$$L^{CLIP}(\theta) = \mathbb{E}_t \left[ \min \left( r_t(\theta)\hat{A}_t, \text{clip}(r_t(\theta), 1 - \epsilon, 1 + \epsilon)\hat{A}_t \right) \right] \tag{4}$$

where $r_t(\theta)$ is the probability ratio, $\hat{A}_t$ is the advantage estimate, and $\epsilon$ is a small constant controlling the clipping range. This approach helps prevent drastic policy updates, ensuring stable training. The PPO implementation improved performance, generating more coherent signature strokes than baseline methods. Effective policy update management helped the agent balance exploration and exploitation, boosting signature quality.

**Trust Region Policy Optimization (TRPO)** Following the implementation of PPO, we explored the TRPO algorithm to enhance the signature generation process. TRPO uses a trust region optimization method that constrains the policy update step by solving the following constrained optimization problem:

$$\max_{\theta} \mathbb{E}_t \left[ \hat{A}_t \right] \quad \text{s.t.} \quad \mathbb{E}_t \left[ \text{KL}(\pi_{\theta_{old}} || \pi_\theta) \right] \leq \delta \tag{5}$$

where $\delta$ is a predefined threshold and KL is the Kullback-Leibler divergence (KLD) (Belov & Armstrong, 2011)between the old and new policies. This constraint significantly improves the stability of the learning process, allowing the SIGN-Agent to produce smoother and more realistic signature strokes. The TRPO's ability to manage the trade-off between exploration and exploitation resulted in refined signature outputs, effectively overcoming limitations observed with earlier methods.

**Advantage Actor-Critic (A2C)** To complete our analysis, we integrated the A2C algorithm, which combines the benefits of both policy and value-based methods. A2C utilizes an advantage function defined as:

$$A(s_t, a_t) = Q(s_t, a_t) - V(s_t) \tag{6}$$

where $Q(s_t, a_t)$ represents the action-value function, and $V(s_t)$ is the state-value function. By leveraging this advantage estimate, A2C improves learning efficiency, allowing the SIGN-Agent to generate signatures with enhanced accuracy and continuity. The incorporation of an advantage estimate reduces variance in the updates, leading to more consistent signature generation performance across different samples. The structured training of A2C, with its synchronous parallel agents, facilitates effective exploration of the action space, further improving the quality of the generated signatures.

The evolution of the SIGN-Agent across the on-policy RL algorithms implemented in this work can be understood through the set of equations presented below. We begin by building on the Bellman equation for the optimal state-value function $V(s)$ in on-policy settings, as shown in Eqn. 7. Here, $(s, s')$ represent the current and consecutive states, and $P$ denotes the environment's transition probability distribution, from which $s'$ is sampled. The reward is represented by $r$, and $\gamma$ is the discount factor.

$$V(s) = \mathbb{E}_{s' \sim P} \left[ r(s) + \gamma V(s') \right] \tag{7}$$

The core component here is the advantage function $A(s, a)$, which measures how much better taking action $a$ in state $s$ is compared to the expected value. The advantage function is approximated using a learned value function $V_\phi(s)$, and the generalized advantage estimation (GAE) is employed to reduce variance in policy updates, as shown in Eqn. 8.

$$A(s, a) = \sum_{t=0}^{T} \left[ r_t + \gamma V_\phi(s_{t+1}) - V_\phi(s_t) \right] \tag{8}$$

With the goal of maximizing the policy performance, PPO optimizes a clipped objective to ensure the updates do not diverge too far from the previous policy. The PPO loss function is defined in Eqn. 9, where $\pi_\theta$ is the current policy, and the clipping parameter $\epsilon$ controls the update size.

$$L^{PPO}(\theta) = \mathbb{E}t \left[ \min \left( \frac{\pi\theta(a_t|s_t)}{\pi_{\theta_{old}}(a_t|s_t)} A_t, \text{clip} \left( \frac{\pi_\theta(a_t|s_t)}{\pi_{\theta_{old}}(a_t|s_t)}, 1 - \epsilon, 1 + \epsilon \right) A_t \right) \right] \tag{9}$$

To handle the policy update smoothly in a trust-region, TRPO uses a constrained optimization approach, ensuring that the KL-divergence between the old and new policies stays below a certain threshold. The TRPO update equation, constrained by KL-divergence, is given by Eqn. 10.

$$\theta' = \arg\max_{\theta} \mathbb{E}t \left[ \frac{\pi\theta(a_t|s_t)}{\pi_{\theta_{\text{old}}}(a_t|s_t)} A_t \right] \quad \text{s.t.} \quad \mathbb{E}t \left[ DKL(\pi_{\theta_{\text{old}}}, \pi_\theta) \right] \leq \delta \tag{10}$$

A2C, as a simpler synchronous version of asynchronous methods, computes policy and value function gradients in parallel over multiple environments. The policy gradient loss for A2C is shown in Eqn. 11, where $\log \pi_\theta(a_t|s_t)$ denotes the log-likelihood of taking action $a_t$ under the current policy.

$$L^{A2C}(\theta) = \mathbb{E}t \left[ \log \pi\theta(a_t|s_t) A_t \right] \tag{11}$$

**Summary of Strengths:** - PPO: Stability and robustness to variability. - TRPO: Smooth updates and precision in trajectory generation. - A2C: Efficiency in learning from diverse trajectories.

The integration of these algorithms allows SIGN-Agent to balance stability, adaptability, and convergence speed in generating user-specific online signatures.

## B APPENDIX B:DETAILS OF THE ALGORITHM

PPO-based Signature Generation algorithm description is provided below:

---
**Algorithm 1** PPO-based Signature Generation
---
1: **Initialize:**
2:     Policy network $\pi_\theta$ and Value network $V_\phi$
3:     Set learning rate $\alpha$, clipping factor $\epsilon$, discount factor $\gamma$,
4:     and max steps $N_{\max}$
5: **for** each iteration **do**
6:     **for** each episode **do**
7:         **Collect Trajectories:**
8:         **for** step $t$ in 1 to $N_{\max}$ **do**
9:             Sample action $a_t$ from policy $\pi_\theta(a_t \mid s_t)$
10:             Execute action $a_t$, observe reward $r_t$ and next state $s_{t+1}$
11:             Store transition $(s_t, a_t, r_t, s_{t+1})$ in memory
12:         **end for**
13:     **end for**
14:     **Compute Advantages:**
15:     **for** each transition in memory **do**
16:         Calculate advantage $\hat{A}_t$ using rewards and value estimates
17:     **end for**
18:     **Update Policy:**
19:     Calculate the surrogate loss using the advantages
20:     Clip the objective to limit policy updates
21:     Perform gradient ascent on the policy network to improve $\theta$
22:     **Update Value Function:**
23:     Compute value loss based on the difference between estimated values and true values
24:     Perform gradient descent on the value network to improve $\phi$
25: **end for**

---

