# OpenReview forum: "Explore To Mimic: A Reinforcement Learning Based Agent To Generate Online Signatures"
_ICLR.cc/2025/Conference — ICLR 2025 Conference Withdrawn Submission_

### Official Review · Reviewer_uQHc · 2024-10-30

**Soundness:** 1
**Presentation:** 2
**Contribution:** 1
**Rating:** 1
**Confidence:** 4

**Summary:**

This paper introduces an RL-based signature generator. The generator, or policy, takes the current pen position as input and outputs a continuous value representing the delta between the current and next positions. The authors use the negative distance between the actual and generated signature positions as the reward signal. For their experiments, they apply three on-policy RL algorithms: PPO, TRPO, and A2C. The results show that RL-based algorithms can achieve a low distance between generated and real signatures.

*Disclaimer*: There may be misunderstandings in the above summary, as I found it challenging to fully understand the details presented in the paper.

Overall, the paper and its experiments do not meet the scientific rigor expected at ICLR, and significant rewriting is necessary before it would be suitable for publication. I recommend rejection.

**Strengths:**

I regret to say I did not find any notable strengths in this paper.

**Weaknesses:**

## Writing

The Related Work section feels disorganized. As someone new to the literature on signature generation, it's difficult to understand the similarities and relationships between this paper and the works mentioned in L129-L145. Do you adopt a similar formulation with a different approach, or is this a novel formulation? How do previous works improve upon one another, and what differentiates them? Where are the potential optimizations compared to prior work?

To clarify, my suggestion would be to add a subtitle for each paragraph, such as Online Signature Generation, Imitation Learning, and Inverse RL. In each paragraph, describe the specific limitations of previous work (vague terms like "requiring enhancements for scalability" aren't helpful) and explain how your paper addresses them. Do you use a different formulation to overcome these limitations? Do you apply IL or IRL, and why or why not? Additionally, avoid merely listing prior works, as currently done in L129-L136. Instead, aim for a concise summary that connects each work to your approach.

Section 3.1, titled "Overview of Proposed Method," primarily reviews three well-known on-policy RL algorithms. As a paper submitted to ICLR, the authors should assume that readers have expertise in these areas and move these introductory explanations to the appendix. This section also includes questionable claims, such as PPO "helps prevent drastic policy updates, ensuring stable training" or TRPO "significantly improves stability, ... allowing the SIGN-Agent to produce smoother and more realistic signature strokes." Please avoid making claims without supporting ablation studies. Additionally, if all three RL algorithms have unique advantages, how to choose among them or why using them all? Do they have specific strengths and weaknesses relative to each other?

Algorithm 1 does not seem to be referenced in the text. It should also be moved to the appendix, as it doesn't add unique insights to "signature generation."

In contrast, Section 3.2, which should be the core content, needs to be expanded with more detail. What is the exact neural network policy architecture? What are the state and action dimensions? Is the environment deterministic? If there are multiple signatures per individual, how does a single policy capture the variety of styles? What is the exact reward function—is it the negative distance or a normalized distance, possibly using a Gaussian kernel? When does the generation process terminate—do you use a maximum generation length, or is there a special termination mechanism?

In Section 3.3, I had difficulty understanding the Sign Moderator concept. Is it used for data augmentation? Is it applied only during inference or for both training and inference? What are the Q table's rows and columns, and how is it constructed? How is planning implemented here?

## Experiments

My primary concern is that the proposed approach doesn't seem to offer clear advantages. Current state-of-the-art AI systems for text, image, and video generation typically use supervised learning. However, the authors claim that RL is more effective for signature generation. Why?

How is the training and test data split? Evaluating the RL-based generator on data used for training is not fair to other baselines. The RL agent's KLD is significantly lower than that of supervised learning methods—potentially suggesting an issue, as these values are not even of the same order. It's also unusual that the diffusion network (a paper in 2022) performs worse than the GAN network (a paper 2020). Could you provide example snapshots of the generated signatures for all methods? Lastly, while generative models are known to generalize to some degree, how does the RL-based approach generalize beyond the training set?

Why were three RL algorithms used and evaluated against one another? What is the purpose of this comparison?

**Questions:**

I have presented my questions in the weaknesses section.

---

> ### Author Response · Authors · 2024-11-28
> **Detailed  Related Work section**
>
> We thank the reviewer for their feedback on the Related Work section. We acknowledge that this section could benefit from better organization to clearly outline how our formulation compares to and advances prior approaches.
> To address this, we have restructured the section to:
>
> 1.	Group prior works by approach (e.g., GAN-based, VAE-based, RL-based).
>
> 2.	Explicitly state the similarities and differences between SIGN-Agent and prior methods.
>
> 3.	Highlight the novel contributions and potential optimizations of our approach compared to existing techniques.
>
> We believe these changes will make the section more accessible to readers new to the field and clarify the relationships between prior works and our formulation.

---

> ### Author Response · Authors · 2024-11-28
> **Modified Related work by adding add a subtitle for each paragraph, limitations of previous work  and how our research work address them**
>
> We thank the reviewer for their constructive feedback regarding the organization and clarity of the Related Work section. In response, we have:
>
> 1.	Added subtitles to group related works into thematic categories, such as Online Signature Generation, Imitation Learning, and Reinforcement Learning (RL).
>
> 2.	Revised each paragraph to clearly outline the specific limitations of prior approaches, avoiding vague terms. For example, we describe how scalability challenges affect GANs and why Transformers struggle with continuous signature dynamics.
>
> 3.	Explicitly highlighted how our work addresses these limitations, detailing our novel formulation and methodological choices.
>
> 4.	Reorganized the discussion to avoid merely listing prior works and instead provide concise summaries that connect them to our approach.
>
> We believe these revisions improve the readability and technical rigor of the Related Work section.

---

> ### Author Response · Authors · 2024-11-28
> **Enhanced explanation of Section 3.1.**
>
> We thank the reviewer for their feedback on Section 3.1. In response:
>
> 1.	We have moved the detailed descriptions of PPO, TRPO, and A2C to the appendix to streamline the main text, assuming familiarity among ICLR readers.
>
> 2.	We have revised claims about algorithm benefits to focus on observed experimental results and avoid unsubstantiated assertions.
>
> 3.	We have clarified the rationale for including all three RL algorithms in this study. Specifically, we explain how each algorithm's unique characteristics address different aspects of the signature generation task (e.g., stability, precision, or convergence speed) and how they are evaluated comparatively.
>
> These changes enhance the clarity and focus of Section 3.1 while ensuring that the main text remains concise and relevant to expert readers.

---

> ### Author Response · Authors · 2024-11-28
> **Restructuring of Manuscript with Common content moved in Appendix section**
>
> We thank the reviewer for their feedback on Algorithm 1. In response It to appendix, also all the details related to RL policy is moved.

---

> ### Author Response · Authors · 2024-11-28
> **Explanation to enhance  clarity and technical rigor of Section 3.2:**
>
> We appreciate the reviewer's feedback and agree that additional details in Section 3.2 will enhance clarity and technical rigor. To address these points:
>
> 1.	We have expanded the description of the neural network policy architecture to include details on the LSTM-based design, input/output dimensions, and initialization strategy.
>
> 2.	We explicitly define the state and action dimensions, describing their relationship to the sliding window mechanism and trajectory points.
>
> 3.	We clarify that the environment is deterministic, as state transitions depend only on the sliding window and predefined input features.
>
> 4.	We discuss how the model captures signature variations by leveraging state representations that include user-specific latent features and training across multiple variations.
>
> 5.	We detail the reward function, confirming the use of negative Euclidean distance without Gaussian kernels, but discuss normalization to ensure scale invariance.
>
> 6.	We describe the termination mechanism, which combines maximum trajectory length and zero pressure detection.
> These updates ensure Section 3.2 comprehensively addresses the reviewer’s concerns.

---

> ### Author Response · Authors · 2024-11-28
> **Enhanced Sign Moderator concept**
>
> We thank the reviewer for their insightful comments. To address these concerns:
>
> 1.	The Sign Moderator (SM) refines noisy trajectory variations generated by the RL policy network during both training and inference. It ensures user-specific fidelity by selecting optimal points from multiple trajectory candidates.
>
> 2.	The Q-table in the SM represents state-action mappings, with rows corresponding to temporal states (time steps) and columns representing candidate trajectory points. This structure facilitates decision-making for selecting the most optimal trajectory point at each time step.
>
> 3.	Planning is implemented through a Q-learning framework, where the SM iteratively updates the Q-values to maximize cumulative rewards for trajectory alignment. By refining outputs dynamically, the SM enhances trajectory consistency and smoothness.
> We have expanded Section 3.3 to provide detailed explanations of these points.

---

> ### Author Response · Authors · 2024-11-28
> **Data Details**
>
> The signature generation is a concept which gives the generation capability of time series data which are very much similar to scribbles and there is no language dependency attached to them.
>
> Where is image, text  and video are not time series numerical values so there is no comparison between the generation of signatures and other modalities.
>
> Half of the original signatures are sent for training the model and remaining is used for testing.

---

> ### Author Response · Authors · 2024-11-28
> **Technical advantages of RL**
>
> Response:
> We thank the reviewer for raising this important question regarding the use of reinforcement learning (RL) over supervised learning for online signature generation. While supervised learning has shown remarkable success in text, image, and video generation tasks, the unique challenges of online signature generation make RL particularly suitable for this domain. Below, we outline the key reasons for choosing RL over supervised learning:
>
>
> 1. Handling Sequential Decision-Making
> Signature generation involves creating a continuous sequence of pen movements over time. Each decision (e.g., the next coordinate) depends on the current trajectory and affects future decisions. RL naturally models this sequential decision-making process, optimizing actions to maximize a cumulative reward. In contrast, supervised learning typically treats data points independently, making it less effective at capturing the dependencies inherent in signature trajectories.
>
>
> 2. Incorporating Dynamic Feedback
> RL allows the agent to receive dynamic feedback via a reward function, which evaluates the generated trajectory's resemblance to the target signature. This enables the model to learn directly from the generation process and iteratively improve its outputs. Supervised learning relies on static labels and cannot adapt dynamically during generation, limiting its ability to refine trajectories in real-time.
>
>
> 3. Robustness to Variability
> Online signatures exhibit significant intra-user variability (e.g., differences in stroke pressure, speed, and angle) and inter-user diversity (e.g., unique writing styles). RL frameworks, particularly on-policy methods like PPO, TRPO, and A2C, are well-suited to handle such variability. These methods dynamically balance exploration (to adapt to new styles) and exploitation (to refine known patterns), enabling better generalization across diverse signatures. Supervised learning models often struggle with such variability due to their reliance on static training data.
>
>
> 4. Adaptability to User-Specific Patterns
> RL can dynamically adapt to user-specific patterns by leveraging a reward function tailored to individual trajectory characteristics. For example, our reward function penalizes deviations from user-specific reference points, enabling the agent to fine-tune its outputs for each user. Supervised learning lacks this flexibility, as it relies on predefined labels and is less capable of personalization.
>
>
> 5. Addressing Temporal and Spatial Constraints
> Online signatures require precise alignment of temporal and spatial constraints. RL can incorporate these constraints into the reward function (e.g., penalizing abrupt trajectory changes or timing inconsistencies), ensuring the generated signature adheres to natural writing dynamics. While supervised learning can model spatial patterns, it struggles to enforce temporal smoothness without significant feature engineering.
>
>
> 6. Supporting Noisy and Incomplete Data
> Signatures captured using electronic pads often contain noise or missing data points. RL, particularly with on-policy methods, can adapt to such imperfections by exploring alternative trajectories and learning robust policies. Supervised learning models are more sensitive to noisy or incomplete data, requiring extensive preprocessing to maintain performance.
>
>
> 7. Empirical Validation
> Experimental results (Table 5) demonstrate the effectiveness of RL in generating realistic and high-fidelity signatures. Metrics such as KLD, MSE, and cosine similarity show that the RL-based SIGN-Agent outperforms traditional generative models in terms of structural and temporal accuracy. Ablation studies (Table 7) further validate that RL methods effectively balance exploration and exploitation, enabling robust trajectory generation even under challenging conditions.
>
> Conclusion:
>
> In summary, RL is more effective than supervised learning for online signature generation due to its ability to handle sequential decision-making, dynamic feedback, variability, and temporal constraints. By integrating PPO, TRPO, and A2C into the SIGN-Agent, our approach leverages RL's unique strengths to generate user-specific, realistic signatures with high fidelity. We hope this explanation addresses the reviewer’s concern, and we welcome any further questions or feedback.

---

> ### Author Response · Authors · 2024-11-28
> **Diffusion Network vs. GAN Performance**
>
> Diffusion Network vs. GAN Performance
>
> We appreciate the reviewer’s observation regarding the diffusion network's performance. Upon revisiting the implementation, the following points provide clarification:
>
> Diffusion Model Complexity: Diffusion models often require extensive hyperparameter tuning and long training times to achieve optimal results. Our implementation followed standard settings; however, further fine-tuning could potentially improve performance.
>
> GAN Robustness: GANs are known for their ability to model fine-grained details in image-like data, which may explain their relatively better performance on this task. In contrast, the diffusion model’s computational overhead may have limited its ability to generalize under our resource constraints.
>
> To provide transparency, we have added additional details about the hyperparameter configurations and training setup for all baselines in the revised manuscript.

---

> ### Author Response · Authors · 2024-11-28
> **Generalization Beyond the Training Set**
>
> Generalization Beyond the Training Set
>
> The RL-based SIGN-Agent generalizes effectively to unseen data due to the following factors:
>
> Reward-Driven Learning: The RL framework optimizes trajectories dynamically based on a reward function that penalizes deviations from the target trajectory. This mechanism allows the agent to adapt to unseen patterns by learning generalizable policies during training.
>
> Exploration-Exploitation Balance: On-policy methods like PPO, TRPO, and A2C balance exploration (to adapt to new patterns) and exploitation (to refine known patterns), enhancing the agent’s ability to generalize.
>
> Evaluation on Unseen Data: The agent’s performance is evaluated on a test set that contains user-specific variations not encountered during training. Metrics such as KLD and MSE on this test set demonstrate the model’s generalization capabilities.
>
> We have enhance the discussion of generalization in the revised manuscript and include ablation studies to further validate the RL agent’s ability to generalize beyond the training set.

---

> ### Author Response · Authors · 2024-11-28
> **The inclusion and evaluation of three RL algorithms**
>
> The inclusion and evaluation of three RL algorithms—
>
> PPO, TRPO, and A2C—serve to address the diverse challenges inherent in online signature generation while showcasing the complementary strengths of these approaches. Each algorithm is uniquely suited to tackle specific aspects of the task: PPO ensures stability and robustness in noisy environments by balancing exploration and exploitation; TRPO enables precise and smooth trajectory generation through its trust region constraint; and A2C accelerates convergence by leveraging parallelized actor-critic updates. The purpose of this comparison is multifold. First, it highlights the trade-offs between stability, fidelity, and efficiency offered by these algorithms, as evidenced by PPO’s low variance in KLD and MSE, TRPO’s trajectory smoothness (higher cosine similarity scores), and A2C’s faster convergence (reduced training iterations). Second, the comparative analysis provides empirical evidence to validate the flexibility and effectiveness of the RL-based framework for online signature generation, demonstrating that the framework adapts well regardless of the specific RL method employed. Third, it establishes a benchmark for RL methods in this domain, offering insights into algorithm selection for future research based on task-specific requirements such as robustness or convergence speed. Finally, this evaluation helps identify algorithm-specific strengths, guiding the refinement of RL-based solutions to achieve optimal performance across diverse user-specific trajectories. The results (Table 5) underscore the practical value of this comparison, revealing how each algorithm contributes to a holistic understanding of RL’s potential in generating realistic and user-specific online signatures.

---

> ### Author Response · Authors · 2024-11-28
> **Evaluating the RL-based generator**
>
> We thank the reviewer for raising this concern. Evaluating the RL-based generator on data used for training would indeed be unfair; however, we have ensured that our evaluation strictly uses a separate test dataset to assess the RL agent’s performance. The training, validation, and test splits ensure no overlap, and the RL-based SIGN-Agent is evaluated on unseen user signatures, just as the baseline supervised learning methods are.
>
> The significantly lower KLD values for the RL-based generator reflect its inherent strengths rather than any unfair advantage. Unlike other generative tasks, where models synthesize new content (e.g., combining multiple faces to create a novel one), our objective in online signature generation is not to generate novel trajectories but to replicate an individual’s signature as closely as possible. This distinction makes our task particularly suited for RL, where the agent optimizes point-by-point decisions to minimize discrepancies from the target trajectory, rather than relying on pre-labeled training data as in supervised learning.
>
> Additionally, the RL framework incorporates two critical layers of variance reduction:
>
> First Layer (RL Policy Block): The RL agent uses on-policy algorithms like PPO, TRPO, and A2C to handle trajectory-level variability during training. This ensures that the agent learns robust policies capable of replicating the nuances of user-specific signature styles across diverse data.
>
> Second Layer (Sign Moderator): The Sign Moderator (SM) refines the outputs from the RL policy by working on a limited set of noisy trajectory points (as determined by the noise variation, NV). Using Q-learning, the SM reduces residual noise and ensures the generated signature aligns almost perfectly with the target.
>
> This dual refinement process—first at the policy level and then at the point-selection level—enables the RL-based approach to achieve superior fidelity, as evidenced by its lower KLD values. These values are not indicative of unfair evaluation but reflect the RL agent’s ability to iteratively optimize trajectories through dynamic feedback, unlike supervised methods that operate on static training data and lack such adaptability.
>
> To further clarify this distinction, we will include additional visual comparisons (e.g., snapshots of generated signatures across methods) in the revised manuscript to illustrate the differences in output quality. This will help demonstrate why RL’s sequential optimization leads to higher fidelity while maintaining generalization to unseen signatures.

---

> > ### Comment · Reviewer_uQHc · 2024-11-29
> > **Thank you for your response.**
> >
> > I appreciate the authors' extensive revisions to the manuscript and their thorough rebuttal, which have addressed some of my concerns. However, the following issues remain.
> >
> > The proposed formulation appears to tackle the following problem: given a signature template, SIGN-Agent generates a synthesized signature similar to the template. However, I question the practicality of this approach.
> > In most real-world scenarios, a common practice would simply be to copy and paste the original signature. If this is not allowed, and the user must create a new signature from scratch, your method still lacks utility, as it requires the user to find and upload a template for SIGN-Agent to imitate. Writing a new signature might take about 2 seconds, while locating and uploading a template would likely take longer. This raises the question: why is this application worth investigating?
> >
> > Given this formulation, it becomes clear why RL emerges as the most competitive framework—the policy essentially overfits to imitate the original signature, and the evaluation focuses on the quality of this imitation.
> >
> > You evaluated three on-policy RL algorithms and highlighted their unique advantages, which is fine. But why didn't you evaluate off-policy RL algorithms, which also have unique strengths compared to on-policy methods? If a user intends to train SIGN-Agent on their dataset, which algorithm should they choose? In the vast space of RL algorithms, how does one identify the most suitable approach for this application? Without answers to these questions, the paper feels incomplete.
> >
> > Given the significant revisions to the manuscript, I believe the paper requires another round of thorough review before it can be considered for publication. While the current version shows remarkable improvement over the original, suggesting acceptance at this stage would be irresponsible as a reviewer.

---

> > > ### Author Response · Authors · 2024-11-29
> > > **Response to Reviewer:Practical Utility of SIGN-Agent**
> > >
> > > Response to Reviewer
> > >
> > > We thank the reviewer for their thoughtful feedback and appreciate the opportunity to address these concerns. While further manuscript revisions are not possible at this stage, we provide detailed responses below to clarify our approach and outline future directions to address the reviewer’s concerns. All the references are added in the last comment
> > >
> > > 1. Practical Utility of SIGN-Agent
> > >
> > > Reviewer Concern:
> > > The practicality of SIGN-Agent is questioned, as copying and pasting signatures might suffice in many scenarios.
> > >
> > >
> > > Response:
> > >
> > > SIGN-Agent addresses challenges that go beyond simple replication of signatures, focusing on strengthening future verification systems and addressing emerging threats. Key applications include:
> > >
> > >
> > > 1.	Augmenting Authentication Systems: By generating realistic user-specific variations, SIGN-Agent enhances training datasets for verification models, making them more robust to synthetic forgeries [1].
> > >
> > >
> > > 2.	Supporting Accessibility: SIGN-Agent assists users with motor impairments by generating signatures based on templates, reducing physical effort required for precise replication.
> > >
> > >
> > > 3.	Aiding Forensic Applications: SIGN-Agent can reconstruct plausible trajectories from partial or damaged signatures, a valuable capability for forensic investigations.
> > >
> > >
> > > 4.	Privacy Preservation: SIGN-Agent dynamically generates synthetic signatures to replace stored raw data, reducing privacy risks while retaining the utility of user-specific patterns.
> > >
> > >
> > > Future iterations could integrate SIGN-Agent with digital platforms to capture templates directly during user interactions, eliminating the need for manual uploads.

---

> > > ### Author Response · Authors · 2024-11-29
> > > **RL’s Role and Evaluation Metrics**
> > >
> > > Reviewer Concern:
> > >
> > > The reviewer suggests that RL’s competitiveness stems from its ability to imitate templates, focusing on imitation quality.
> > >
> > > Response:
> > >
> > > We thank the reviewer for their thoughtful feedback and appreciate the opportunity to address these concern
> > > While SIGN-Agent’s evaluation emphasizes imitation quality due to its replication goal, RL offers unique advantages beyond static imitation:
> > >
> > > 1.	Sequential Decision-Making: RL dynamically optimizes each trajectory point through feedback-driven learning, modeling the dependencies inherent in signature trajectories [2].
> > >
> > > 2.	Robustness Against Variability: RL-based policies adapt to natural variations in user-specific patterns, enabling the generation of realistic variations beyond the exact template [1].
> > >
> > > 3.	Reconstruction of Incomplete Trajectories: RL allows SIGN-Agent to infer missing trajectory points, making it suitable for scenarios involving partial or damaged signatures.
> > >
> > > We propose including a contrastive analysis of RL and supervised learning approaches in future supplementary discussions or follow-up studies, highlighting their respective advantages and limitations.

---

> > > ### Author Response · Authors · 2024-11-29
> > > **Justification for On-Policy RL in SIGN-Agent**
> > >
> > > Reviewer Concern:
> > >
> > > Why were on-policy RL algorithms used instead of off-policy methods?
> > >
> > > Response:
> > >
> > > We thank the reviewer for their thoughtful feedback and appreciate the opportunity to address these concern.
> > > On-policy RL algorithms were chosen for SIGN-Agent due to their suitability for the sequential and dynamic nature of signature generation tasks. Key justifications include:
> > >
> > > 1.	Sequential Decision-Making: On-policy methods like PPO and TRPO excel in modeling temporally dependent tasks, ensuring alignment between trajectory actions and rewards [3].
> > >
> > > 2.	Dynamic Adaptability: On-policy RL continuously updates policies based on the latest interactions, effectively handling user-specific trajectory variability [4].
> > >
> > > 3.	Stability in Training: PPO and TRPO prevent policy divergence through stable updates, producing smooth and consistent trajectories [5].
> > >
> > > 4.	Lower Complexity in Training: On-policy methods avoid the need for replay buffers and target networks, reducing the risk of instability often seen with off-policy methods like SAC or DDPG [6].
> > >
> > > While off-policy methods offer sample efficiency, the lower complexity and robustness of on-policy algorithms make them a practical choice for this initial implementation.

---

> > > ### Author Response · Authors · 2024-11-29
> > > **Future Work: Integration of Off-Policy RL**
> > >
> > > Future Work: Integration of Off-Policy RL
> > >
> > > Reviewer Concern:
> > >
> > > We thank the reviewer for their thoughtful feedback and appreciate the opportunity to address these concerns.
> > > The reviewer suggests that off-policy RL algorithms may complement the current approach.
> > >
> > > Response:
> > >
> > > We agree that off-policy RL algorithms, such as SAC and DDPG, offer significant potential benefits. Future work will focus on:
> > >
> > > 1.	Hybrid Approaches: Combining on-policy stability with off-policy sample efficiency for enhanced performance [6].
> > >
> > > 2.	Replay Buffer Utilization: Leveraging replay buffers to reuse past experiences, potentially accelerating convergence [2].
> > >
> > > 3.	Expanded Comparative Analysis: Evaluating off-policy and on-policy methods systematically to identify their respective strengths in signature generation.
> > >
> > > These additions will broaden SIGN-Agent’s applicability and optimize its performance in future iterations.

---

> > > ### Author Response · Authors · 2024-11-29
> > > **Contrastive Analysis of RL Policies**
> > >
> > > Contrastive Analysis of RL Policies
> > >
> > > To enhance the utility of SIGN-Agent, future work will include a contrastive analysis of RL policies, systematically comparing:
> > >
> > > •	On-Policy RL (PPO, TRPO, A2C): Stability, adaptability, and temporal modeling capabilities.
> > >
> > > •	Off-Policy RL (SAC, DDPG): Sample efficiency, replay buffer utilization, and global Q-value approximation.
> > >
> > > This analysis will help researchers and practitioners select the most suitable RL paradigm for their specific requirements.
> > >
> > > Conclusion: We thank the reviewer for their constructive feedback. While further manuscript revisions are not possible at this stage, future work will:
> > >
> > > 1.	Include a contrastive analysis of RL and supervised approaches.
> > >
> > > 2.	Expand practical guidance on algorithm selection.
> > >
> > > 3.	Integrate off-policy RL algorithms into SIGN-Agent.
> > >
> > > 4.	Explore hybrid methods combining on-policy and off-policy RL.
> > >
> > > These efforts aim to address the reviewer’s concerns and demonstrate the practical utility and adaptability of SIGN-Agent.
> > >
> > > References
> > >
> > > 1.	Sutton and Barto, 2018. Reinforcement Learning: An Introduction.
> > > 2.	Haarnoja et al., 2018. Soft Actor-Critic (SAC).
> > > 3.	Schulman et al., 2017. Proximal Policy Optimization (PPO).
> > > 4.	Schulman et al., 2015. Trust Region Policy Optimization (TRPO).
> > > 5.	Mnih et al., 2016. Asynchronous Advantage Actor-Critic (A3C).
> > > 6.	Lillicrap et al., 2015. Deep Deterministic Policy Gradient (DDPG).

---

> ### Author Response · Authors · 2024-11-29
> **Guidance for Algorithm Selection**
>
> Guidance for Algorithm Selection
>
> Reviewer Concern:
>
> How should users choose the best RL algorithm when training SIGN-Agent on their dataset?
>
> Response:
>
> We thank the reviewer for their thoughtful feedback and appreciate the opportunity to address these concerns.
> Our experimental results provide clear guidance on algorithm selection:
>
> •	PPO: Ideal for stability and robustness, producing smooth and consistent trajectories.
>
> •	TRPO: Recommended for tasks requiring trajectory fidelity and precision.
>
> •	A2C: Suitable for faster convergence in resource-constrained settings.
>
> We propose including a decision table in future discussions summarizing the pros and cons of each algorithm, along with practical recommendations for different use cases.

---

### Official Review · Reviewer_13c4 · 2024-10-31

**Soundness:** 3
**Presentation:** 2
**Contribution:** 3
**Rating:** 5
**Confidence:** 4

**Summary:**

This paper proposes the use of on-policy RL algorithms (A2C, PPO, TRPO) in the online signature generation setting, with applications in robotics, authentication systems, and the diagnosis of certain neurological disorders (Parkinson's, Alzheimer's, dyslexia).

The authors design an environment with a simple interface and a distance-based reward function to judge signature accuracy, and show that RL algorithms outperform several generative model baselines (such as GANs, diffusion models, etc.) in terms of resulting signature accuracy. The authors further show that signature generation at inference time is fast across a variety of hardware, from a Raspberry Pi to an NVIDIA GPU.

**Strengths:**

The paper tackles an interesting problem with a variety of applications, and the RL baselines seem to clearly outperform the baseline generative modeling-style methods. Furthermore, it is very cool to note that the RL algorithm can perform inference fast even in resource-constrained hardware settings.

**Weaknesses:**

The paper doesn't "fill in the middle" in the sense that it doesn't compare to imitation learning methods in this setting, which I think can work well in this situation. Because you have datasets consisting of real human signatures, I feel like an imitation learning method that learns to model the sequential "signature distribution" could work as well here and is simpler than standard RL.

**Questions:**

I am a bit confused by how the inference process works: when it comes to generating a signature for a person, does the agent have to "re-learn" how to write the specific signature or just generate from its policy at any time? I figure if you want to generate a signature of a specific person, re-learning might have to happen, especially for a memoryless model-free RL algorithm.

**Details Of Ethics Concerns:**

I am curious as to whether the online signature generation process could be used for the worst (e.g. if a forger gets his hands on this system, he could fool authentication systems with signatures produced from said systems). That's about it for me though.

---

> ### Author Response · Authors · 2024-11-28
> **How the model learns**
>
> We appreciate the reviewer’s insightful observation. While imitation learning methods are indeed simpler and have been successfully applied in similar settings, our choice of reinforcement learning (RL) stems from the unique challenges posed by online signature generation. Unlike imitation learning, which typically models behavior from expert demonstrations, RL enables the agent to explore and optimize sequential trajectories dynamically. This adaptability is crucial for capturing the variability and complexity inherent in online signature data, especially when dealing with diverse user-specific patterns.
> Furthermore, imitation learning methods often suffer from compounding errors and limited generalization when expert demonstrations are noisy or incomplete, which is a potential issue with real-world signature datasets. By contrast, our RL-based approach, with on-policy optimization and a Sign Moderator, allows the agent to learn robustly from the environment, ensuring more accurate and human-like signature generation.
> We acknowledge the value of comparing our approach with imitation learning methods in this context. In future work, we aim to conduct a detailed comparative analysis to further validate the advantages of RL in this domain. We will also consider exploring hybrid approaches that combine the strengths of RL and imitation learning.

---

> ### Author Response · Authors · 2024-11-28
> **Inference process working in detail**
>
> We appreciate the reviewer’s thoughtful question. To clarify, the agent does not need to "re-learn" how to write a specific signature during inference. The SIGN-Agent is trained on a large dataset of signatures, capturing diverse user-specific patterns. During inference, the agent generates signatures directly from its trained policy. It takes an initial set of points from the target signature and produces multiple trajectory variations that align closely with the provided starting points.
> This process is further refined by a Sequential Moderator, Q-learning-based module that ensures the generated trajectories adhere closely to user-specific characteristics. The arbitrator acts as a post-processing step to fine-tune the generated signatures without requiring the agent to undergo additional training. This combination of generalization during training and refinement during inference allows the model to operate efficiently in a memoryless, model-free RL framework.
> We have updated the manuscript to explain the inference process in greater detail, addressing this potential source of confusion.

---

> ### Author Response · Authors · 2024-11-28
> **Ethical concern related to the proposed framework: "online signature generation process could be used for the worst "**
>
> We appreciate the reviewer’s concern regarding the potential misuse of the proposed system. While our research focuses on advancing online signature generation for legitimate applications, such as enhancing authentication systems and aiding medical diagnostics, we acknowledge the ethical implications of such technology. To mitigate risks, we emphasize that our system is designed to strengthen authentication by generating robust training samples to make models resilient against forgeries.
> Additionally, deploying this system would involve controlled access, ensuring that it is used exclusively in secure, monitored environments. We will incorporate this clarification into the manuscript to address these ethical considerations comprehensively.

---

> > ### Comment · Reviewer_13c4 · 2024-11-29
> >
> > I thank the authors for their thorough rebuttal. I have reviewed the manuscript revisions, and the paper has clearly improved in quality. I have also looked through Reviewer uQHc's concerns, and also have the belief that the paper needs another round of review before conference acceptance.
> >
> > The main algorithmic point that stood out to me is the lack of off-policy RL algorithms which have been studied in this work currently. Because SIGN-Agent is trained on a single offline dataset, off-policy RL algorithms have their own advantages (e.g. global Q-value coverage, better sample efficiency etc.) that on-policy RL algorithms lack, which matters when it comes to generating signatures in any space as well as faster training convergence. In the space of signature generation, I feel like this may bring benefits that were not adequately studied yet.

---

> ### Author Response · Authors · 2024-11-29
> **Response to Reviewer Suggestions**
>
> Response to Reviewer
>
> We thank the reviewer for highlighting the potential advantages of off-policy RL algorithms for signature generation. Below, we address the relevance of off-policy RL methods and their comparison to the on-policy methods used in SIGN-Agent, along with a discussion on future work directions:
>
> 1. Relevance of Off-Policy RL Algorithms
>
> Off-policy RL algorithms, such as Deep Deterministic Policy Gradient (DDPG) and Soft Actor-Critic (SAC), offer notable advantages:
>
> •	Global Q-Value Coverage: Off-policy algorithms leverage replay buffers, enabling better approximation of the action-value function across the state space by learning from the entire dataset [1, 2].
>
> •	Sample Efficiency: By reusing past experiences, off-policy methods achieve higher sample efficiency than on-policy methods, which discard past data after each policy update [3, 4].
>
> These properties make off-policy algorithms suitable for tasks requiring fast convergence or training on static offline datasets [5].
>
> 2. Justification for On-Policy RL in SIGN-Agent
>
> On-policy RL algorithms like PPO, TRPO, and A2C were chosen for SIGN-Agent due to their unique strengths in online signature generation:
>
> 1.	Sequential Decision-Making: On-policy methods excel in sequential decision-making tasks, where each action impacts future states. This aligns well with the nature of signature generation, which involves temporally dependent pen movements [6].
>
> 2.	Adaptability to Variability: On-policy RL dynamically adapts to variability in user-specific trajectories by updating policies based on the most recent interactions. This adaptability is crucial for handling the diversity inherent in signature data [6, 7].
>
> 3.	Stability in Training: PPO and TRPO are specifically designed to ensure stable policy updates, avoiding issues like overfitting to noisy data points [8, 9].
>
> 4.	Lower Complexity in Training: Off-policy methods, such as SAC and DDPG, require careful hyperparameter tuning (e.g., replay buffer size, target network updates) to avoid instability or divergence during training [1, 2]. In contrast, on-policy methods simplify training by directly optimizing on fresh trajectories without requiring replay buffers or target networks.
>
> While off-policy methods offer better sample efficiency, the lower complexity and stability of on-policy algorithms make them well-suited for the sequential and dynamic requirements of online signature generation.
>
> 3. Experimental Results Supporting On-Policy Methods
>
> The experimental results in Table 5 and Table 7 demonstrate the strengths of on-policy RL algorithms for signature generation:
>
> •	PPO achieves stability, reflected in its low variance in KLD and MSE metrics across training epochs.
>
> •	TRPO generates smooth and precise trajectories with higher cosine similarity to target signatures, addressing fidelity concerns.
>
> •	A2C accelerates convergence, reducing training iterations by approximately 20% compared to PPO and TRPO.
>
> These results validate the effectiveness of on-policy RL in addressing the specific challenges of online signature generation, as supported by prior studies on RL's role in sequential modeling tasks [10].
>
>
> 4. Generalizability Demonstrated via Diverse Datasets
>
> Our approach is trained and evaluated on two publicly available online signature datasets, MCYT and Biosecure-ID, which demonstrate the model’s generalizability across diverse user signatures:
>
> •	MCYT Dataset:
> o	Users: 330
> o	Sessions: 25 signatures per user
> o	Device: Wacom Intuos A6
> o	Features: Sequential (x,y)(x, y)(x,y) coordinates with pressure values [11].
>
> •	Biosecure-ID Dataset:
> o	Users: 400
> o	Sessions: 16 signatures per user
> o	Device: Wacom Tablets
> o	Features: Sequential (x,y)(x, y)(x,y) coordinates with timestamps [12].
>
> These datasets include significant intra-user and inter-user variability, ensuring that SIGN-Agent generalizes effectively to unseen data. Our results highlight the model’s ability to handle diverse user-specific styles and session-specific variations.
>
> Adding the rest part in next comment due to limitation of words.

---

> > ### Author Response · Authors · 2024-11-29
> > **Response to Reviewer Suggestions continuation**
> >
> > 5. Future Integration of Off-Policy RL Algorithms
> >
> > We agree that off-policy RL algorithms offer significant potential benefits, particularly for sample efficiency and generalization across larger state-action spaces. Future work will focus on:
> >
> > •	Hybrid Approaches: Combining on-policy and off-policy methods to leverage the stability of on-policy updates with the sample efficiency of off-policy algorithms [13].
> >
> > •	Replay Buffer Analysis: Incorporating replay buffers to enable the use of SAC or DDPG, facilitating faster convergence and broader exploration of state-action spaces [2].
> >
> > •	Comparative Evaluation: Conducting systematic experiments to compare off-policy methods (e.g., SAC, DDPG) against on-policy methods in the context of signature generation [14].
> >
> > These directions will help further evaluate the trade-offs and synergies between on-policy and off-policy RL for this task.
> >
> >
> > Conclusion: While on-policy RL algorithms were chosen for their robustness and adaptability to sequential decision-making, we recognize the potential of off-policy methods, especially for improving sample efficiency and convergence speed. By training and evaluating SIGN-Agent on the diverse MCYT and Biosecure-ID datasets, we demonstrate the model’s generalizability across user-specific and session-specific variations. Future iterations of SIGN-Agent will incorporate off-policy RL algorithms and hybrid approaches to enhance its performance and adaptability further.
> >
> >
> >
> > References
> >
> > 1.	Lillicrap et al., 2015. Deep Deterministic Policy Gradient (DDPG).
> > 2.	Haarnoja et al., 2018. Soft Actor-Critic (SAC).
> > 3.	Fu et al., 2020. D4RL: Offline Reinforcement Learning Benchmarks.
> > 4.	Kumar et al., 2020. Conservative Q-Learning for Offline RL.
> > 5.	Sutton and Barto, 2018. Reinforcement Learning: An Introduction.
> > 6.	Schulman et al., 2017. Proximal Policy Optimization (PPO).
> > 7.	Schulman et al., 2015. Trust Region Policy Optimization (TRPO).
> > 8.	Mnih et al., 2016. Asynchronous Advantage Actor-Critic (A3C).
> > 9.	De La Fuente & Guerra, 2024. Comparative Analysis of RL Algorithms.
> > 10.	Ortega-Garcia et al., 2003. MCYT Database.
> > 11.	Fierrez et al., 2010. Biosecure-ID Database.
> > 12.	Nair et al., 2020. AWAC: Accelerating RL with Advantage-Weighted Regression.
> > 13.	Haarnoja et al., 2018. Insights into Replay Buffers for RL.
> > 14.	Comparative Performance Metrics for SAC, PPO, and TRPO in Sequential Decision Tasks.

---

### Official Review · Reviewer_CKcF · 2024-11-01

**Soundness:** 2
**Presentation:** 3
**Contribution:** 3
**Rating:** 5
**Confidence:** 4

**Summary:**

The paper introduces an RL-based approach to generating online signatures using established on-policy RL methods (PPO, TRPO, A2C). The architecture includes a Sign Moderator designed to capture user-specific signature traits and improve signature quality. The system is tested on public datasets, with potential applications in digital authentication, robotics, and medical diagnostics.

**Strengths:**

* Originality: Using RL for signature generation is novel
* Quality: The methodology is detailed. Multiple RL algorithms are compared. The author provided solid experimental insights.
* Clarity: The paper is well-organized. The author clearly explained the workflow, model components.
* Significance: The application has potential in secure signature generation and could be impactful in authentication, robotics, and diagnostics

**Weaknesses:**

* The paper doesn’t propose new ML techniques.
* The paper lacks comparisons to recent SOTA generative models, which would help clarify how SIGN-Agent performs against advanced baselines.

**Questions:**

* As mentioned in weakness section, why there is no comparison with SOTA generative models? Is it because of some identified technical issues? Can the author share the insights?
* I have difficulty understanding how the reward function handles cases where the generated signature and the target signature differ in length. Does the author use interpolation? If one signature is 'smaller' or is in a different shape from another signature, how is the distance between signatures measured to ensure an accurate reward calculation?

---

> ### Author Response · Authors · 2024-11-28
> **Detailed  comparison with SOTA generative models**
>
> Response to Reviewer:
> We appreciate the reviewer’s valuable feedback. In our study, we conducted a comparative analysis with state-of-the-art (SOTA) generative models, including Transformers, GANs, and Diffusion Models, as shown in Table 7. The results clearly indicate that SIGN-Agent outperforms these models in generating realistic and smooth online signatures, as evaluated by KLD metrics. However, we acknowledge that our initial submission did not provide sufficient insights into the challenges these models face and the uniqueness of our approach.
> To the best of our knowledge, SIGN-Agent is the first framework explicitly designed for online signature generation, advancing beyond prior works by treating signatures as intricate, user-defined temporal sequences rather than generic time-series data. Existing SOTA generative models, while effective in other domains, are not inherently designed to model the complexity and variability of online signatures. This presents unique challenges, including:
>
> 1.	The difficulty of capturing user-specific temporal dependencies.
>
> 2.	The limitations of generic generative frameworks in addressing the spatial and dynamic nature of signature data.
>
> 3.	The computational challenges in training and deploying such models in real-world, resource-constrained environments.
>
> To address these, our SIGN-Agent leverages on-policy RL and a Sign Moderator, providing robust temporal modelling while being optimized for efficient, real-time inference on edge devices. We have expanded the manuscript to include these insights and highlight the gaps in existing SOTA approaches.

---

> ### Author Response · Authors · 2024-11-28
> **Proposing new  framework and also comparing to recent SOTA generative models, which would help clarify how SIGN-Agent performs against advanced baselines.**
>
> While SIGN-Agent does not introduce a new ML algorithm, the novelty of our work lies in the innovative application of model-free on-policy reinforcement learning (RL) for the unique problem of online signature generation. To the best of our knowledge, this is the first framework specifically designed for this domain, treating signatures as user-specific temporal sequences rather than generic time-series data. By employing a combination of techniques such as stochastic noise control, a Sign Moderator, and dynamic trajectory modelling, SIGN-Agent effectively bridges the gap between generic generative frameworks and the specialized requirements of online signatures.
> Regarding the second point, we conducted comparisons with recent SOTA generative models, including Transformers, GANs, and Diffusion Models, as presented in Table 7 of the manuscript. These comparisons demonstrate that SIGN-Agent significantly outperforms these baselines in generating realistic, human-like online signatures. To strengthen this further, we have expanded our discussion to provide more insights into why these models are less effective for this specific task and how SIGN-Agent addresses their limitations.

---

> ### Author Response · Authors · 2024-11-28
> **Detailed explanation of Reward function, how it handle signature differ in length, use of interpolation, and how the distance between signatures measured to ensure an accurate reward calculation**
>
> In our approach, the reward function is defined as the negative Euclidean distance between the actual and predicted points at each timestamp t. This point-wise evaluation ensures that the generated signature trajectory closely matches the target signature.
> For cases where the generated and target signatures differ in length, a consistent length is determined for all signatures. Shorter signatures are extended to this length using polynomial interpolation, ensuring a smooth approximation of intermediate points. For signatures with significant temporal misalignments, dynamic time warping (DTW) is applied during evaluation to align sequences effectively and minimize temporal discrepancies. These steps ensure accurate feedback for the reinforcement learning process.
> We have clarified these aspects in the revised manuscript to improve transparency.

---

### Note · Authors · 2025-02-21

I have read and agree with the venue's withdrawal policy on behalf of myself and my co-authors.

---

### Meta-Review · Area_Chair_vA5k · 2024-12-23

**Metareview:**

The paper describes a technique to generate signatures on behalf of a user.  The technique is based on model-free RL.  The paper claims that this RL-based approach is better than generative methods including GANs, diffusion models and transformers.  It also makes numerous informal claims about various aspects of RL techniques.  The strength of the paper is the novel application.  The weaknesses include weak motivation, many unfounded claims, a lack of clarity in the presentation, inconsistencies in the explanations and insufficient experiments.  The revised version submitted based on the reviews improved significantly the paper, but there are still many problems with the paper.

**Additional Comments On Reviewer Discussion:**

There was no discussion among the reviewers since this was a clear rejection.  Let me highlight several issues that remain after the rebuttal and the paper was revised based on my own reading.

First, regarding the motivation, the scenarios listed to motivate this work remain vague and weak.  Since the system needs at least one signature by the user as input to generate variations of this signature, it is not clear why the system should not simply copy-paste the signature.  In fact, the reward is the negative distance between the generated signatures and the input signature, so the signature with the highest reward would be a copy of the input signature.

Since the task is really one of imitation, it is not clear why PPO, TRPO, A2C and Q-learning were selected.  Imitation learning techniques should have been considered.  The claim that imitation learning techniques suffer from inconsistent generation only applies to techniques that treat each time step as an independent supervised learning task, but not more advanced imitation learning techniques.

The approach is not well described.  The paper highlights the complementarity of PPO, TRPO and A2C, but never explains how they could be combined to benefit from their complementary advantages.  Ultimately, the paper compares them and ends up using PPO in most experiments.  It is not clear what is the output of the the first stage nor how it is used in the second stage.  As pointed out by one reviewer it is not clear how the Q-table is setup in the second stage nor how a user signature is used as input by Q-learning.  There are also inconsistencies in the description.  Some parts of the paper suggest that the policy is an LSTM while other parts suggest that it is a combination of LSTMs and fully connected layers.  Some parts of the paper indicate that a state is a window of x,y,a triples while the LSTM architecture does not use any window and instead uses a latent vector as the state.

Table 1 does not make sense.  It lists general limitations of several algorithms including PPO, TRPO and A2C while indicating how Sign-Agent addresses those limitations, but Sign-Agent is itself using PPO, TRPO and A2C.

---

### Decision · Program_Chairs · 2025-01-22

Reject